# Public health emergency operations centres in Africa: a cross-sectional study assessing the implementation status of core components and areas for improvement, December 2021

Senait Tekeste Fekadu ,[1] Abrham Lilay Gebrewahid ,[1] Wessam Mankoula,[2] Womi Eteng,[2] Virgil Lokossou,[3] Yan Kawe,[1] Ali Abdullah,[4] L Jian,[5] Mathew Tut M. Kol,[2] Menchion Chuck Wilton,[6] Emily Rosenfeld,[6] Valerie Nkamgang Bemo,[7] Emily Collard,[8] Liz McGinley,[8] Ariane Halm,[9] Merawi Aragaw,[2] Ishata Nannie Conteh ,[1] Fiona Braka,[1] Abdou Salam Gueye[1]

**Correspondence to**
Senait Tekeste Fekadu;
tekestes@who.int

## ABSTRACT

**Objective** To assess implementation status of public health emergency operations centres (PHEOCs) in all countries in Africa.

**Design** Cross-sectional.

**Setting** Fifty-four national PHEOC focal points in Africa responded to an online survey between May and November 2021. Included variables aimed to assess capacities for each of the four PHEOC core components. To assess the PHEOCs' functionality, criteria were defined from among the collected variables by expert consensus based on PHEOC operations' prioritisation. We report results of the descriptive analysis, including frequencies of proportions.

**Results** A total of 51 (93%) African countries responded to the survey. Among these, 41 (80%) have established a PHEOC. Twelve (29%) of these met 80% or more of the minimum requirements and were classified as fully functional. Twelve (29%) and 17 (41%) PHEOCs that met 60%–79% and below 60% the minimum requirements were classified as functional and partially functional, respectively.

**Conclusions** Countries in Africa made considerable progress in setting up and improving functioning of PHEOCs. One-third of the responding countries with a PHEOC have one fulfilling at least 80% of the minimum requirements to operate the critical emergency functions. There are still several African countries that either do not have a PHEOC or whose PHEOCs only partially meet these minimal requirements. This calls for significant collaboration across all stakeholders to establish functional PHEOCs in Africa.

## INTRODUCTION

Public health threats, for example, stemming from natural disasters, and consequent public health emergencies (PHEs) continue to be a major concern for African countries. In Africa, over 100 PHEs are recorded annually, 80% or more are caused by infectious diseases.[1] These include emerging and re-emerging diseases, such as COVID-19, monkeypox, Ebola virus disease (EVD), polio, cholera, Rift Valley fever, Crimean-Congo haemorrhagic fever and yellow fever.[2]

Since the adoption of the International Health Regulations (IHR) 2005, WHO Member States (MS) established systems to minimise the effects of PHEs by improving preparedness and response capabilities.[3–5] However, recent PHEs revealed important gaps in IHR core capacities. Previous experiences, particularly the EVD outbreak in West Africa in 2014–2016, revealed critical gaps in preparing for and responding to PHEs. The governments of Guinea, Liberia and Sierra Leone have acknowledged that poor coordination of efforts and weak health systems hampered the effectiveness of the 2014–2016

EVD outbreak response.[6] Improving preparedness and response capacities of MS as required by IHR (2005) which requires State Parties to develop, strengthen and maintain their capacity to effectively respond to PHEs was one of the recommendations made following this EVD outbreak.[6]

A functional public health emergency operations centre (PHEOC) is a crucial component in meeting the IHR (2005) minimum capabilities and the need to establish a functional PHEOC has been covered as one of the key thematic areas in the joint external evaluation (JEE) developed to help MS assess their IHR-related capacities to prevent, detect and respond to public health threats.[4] A PHEOC is a hub for effective coordination of information and resources during the management of PHEs.[7 8] The PHEOC employs an incident management system (IMS), an emergency management structure with sets of procedures and protocols to provide a coordinated approach for all types of PHEs. However, only after a PHEOC has been thoroughly implemented and continuously strengthened can it serve as an effective platform for managing and coordinating information and resources among the multidisciplinary and multisectoral agencies, governments, organisations, and other stakeholders involved in PHEs preparedness and response.

After the 2014–2016 EVD outbreak in West Africa, WHO Regional Office for Africa (WHO AFRO), Africa Centres for Disease Control and Prevention (Africa CDC), West African Health Organization (WAHO) and other partners heightened the support to MS to establish and operationalise PHEOCs as the top priority. As part of the support, WHO published an evidence-based Framework for a PHEOC in 2015.[7] In the same year, WHO AFRO officially launched a regional emergency operations centre network to collaborate with MS and key partners to assist MS in establishing functional PHEOCs and to promote timely exchange of experiences, best practices and information.[9] Furthermore, the collaboration between WHO AFRO, Africa CDC and US Centers for Disease Control and Prevention (US CDC) since 2018, was instrumental in the development of an easy-to-use handbook for PHEOC operations and management and a PHEOC legal framework guide which provides technical guidance to MS on the development of appropriate policies that authorise PHEOC establishment and full functionality.[4 5]

The PHEOC framework highlights that a fully functional PHEOC is achieved and maintained by putting in place four key components: policies, plans and procedures; information system and data standard; skilled human resources; and communication technology and physical infrastructure. Africa has made considerable strides in establishing PHEOCs by putting in place these key components to strengthen emergency management. Findings from regional-level meetings on PHEOC implementation and other activities and exchanges, including JEEs, revealed that there are remaining challenges in establishing functional PHEOCs in the continent. Until now, no detailed assessment on the status of

PHEOC establishment and functionality in Africa was available. Therefore, WHO in partnership with Africa CDC, WAHO, US CDC, UK Health Security Agency, Bill & Melinda Gates Foundation and Robert Koch Institute conducted a survey to assess the implementation status of core components required to having functional PHEOCs in Africa. The main question to be answered by this survey was 'How are the PHEOCs performing in meeting the minimum requirements of the core components for coordinating functional PHE response in the Africa Region'? The specific objectives were to assess the general implementation status of the four PHEOC core components' parameters, and to evaluate the fulfilment status of each core components' minimum requirements.

The survey further intended to answer the following questions:

► What are the capacities of the PHEOCs regarding communication technology and physical infrastructure?
► How are the key PHEOC policies, plans and procedures for emergency management (eg, legal framework, multihazard plan, PHEOC plan/handbook, hazard-specific plans, functional plans, SOPs) developed and implemented?

What are the PHEOCs capacities in terms of routine and surge emergency management workforce?

The findings of this survey could serve as a resource for planning considerations, generate evidence to contextualise minimum and core requirements for functional PHEOCs, and contribute to policy recommendations.

## METHODS
### Data collection
A standardised self-assessment tool was developed based on Annex 9 of the PHEOC framework to determine the PHEOC minimum requirements (Checklist for Implementing a PHEOC[7]). It encompasses 31 questions resulting in variables for the analysis across the four defined PHEOC core components to assess the implementation status and existing capacities. Each question allowed for a yes or no answer as well as a comment section to provide additional information if desired. The questionnaire was conceived in Kobo Toolbox and finalised in May 2021, the survey link was shared with WHO country offices and then shared with PHEOC focal points of all countries, and these could complete and submit their responses online. The online survey questionnaire was shared with 54 MS in Africa (47 MS from WHO African Region) and these responded based on their own first-hand experience. Data were collected between May and November 2021. In addition, the Regional Office set up a one-on-one discussion with focal points from selected countries to verify and ensure completeness their responses.

To allow for a structure, systematic approach and comparison, different parameters were selected under each PHEOC core component (number of variables per

**Table 1** Assessed variables by PHEOC core component in Africa, 2021*

| Core components of a PHEOC | Variables assessed (No) |
|---|---|
| PHEOC policies, plans and procedures | 12 |
| Human resources, training and simulation exercises | 7 |
| Information management and data standards | 4 |
| Communication technology and physical infrastructure | 8 |
| Total | 31 |

*Results are only presented for selected parameters in the next section.
PHEOC, public health emergency operations centre.

component, table 1) and used to assess the operationalisation of PHEOCs across the region.

## Data analysis

In addition to all variables collected for the general PHEOC core components status analysis, the minimum required PHEOC capabilities for running critical emergency response functions were selected from these variables included in the survey tool. This selection was done based on the WHO framework for a PHEOC and expert consultation. Members of the above-mentioned group of partners committed to strengthening PHEOC capacities in Africa prioritised and selected specific parameters within each core component based on PHEOC operations and consensus. As a result, 19 of the 31 (table 2) parameters from the four PHEOC core components were prioritised and classified the PHEOCs into three categories for planning purpose: PHEOCs meeting 80% and above of the minimum requirements (n≥15 of the 19) were labelled as 'fully functional', PHEOCs meeting 60%–79% (11–14 of the 19) as 'functional', and PHEOCs that met below 60% (≤10 of the 19) labelled as 'partially functional'. The survey data were extracted from Kobo Toolbox into Excel used to analyse the data. Simple frequencies and proportions were calculated for each of the four core components' variables and the above-mentioned scale used for a general PHEOC classification into one of the three categories regarding their functionality. The same weight was given to each parameter assess PHEOC functionality. For selected variables, additional calculations were made for those MS part of the WHO African Region (subgroup).

## Patient and public involvement
None.

## RESULTS
A total of 51 (93%) MS PHEOC focal points in the African continent responded to the survey. Forty-one (80%) reported that they have an established PHEOC at the national level and 21 of these MS also had a PHEOC at the subnational level. Forty-one of the 47 WHO African

**Table 2** Parameters used to classify the PHEOC core components minimum requirements fulfilment level in Africa, 2021

| PHEOC policy, plans and procedures | Human resource, training and simulation exercise | Information management and data standards | Communication technology and physical infrastructure |
|---|---|---|---|
| The legal framework authorises a PHEOC at the national level | PHEOC has the minimum requirements for routine staff (PHEOC manager and key IMS) | A direct link to the national surveillance structure exists | Dedicated PHEOC facility with adequate workstations for the key IMS functions |
| Legal frameworks include governance structure, core functions and authority | Staff trained in PHE preparedness and response | Essential data systematically flow to the PHEOC from relevant sectors | Availability of internet access for all workstations and meeting rooms |
| Relationships, before, during and after a PHE between stakeholders defined | PHEOC has a comprehensive exercise programme involving more than one exercise per year | Able to collect and manage operational information to inform leadership | Availability of electricity with backup power |
| A policy group to provide strategic guidance established | Staff are routinely trained on guides, plans and through simulation to validate competencies | PHEOC uses digital solutions to process its information | |
| Handbook for PHEOC operations and management is in place | Contact roster of trained personnel to fill IMS positions when PHEOC activated is available | | |
| The multihazard response plan is approved | | | |
| A business continuity plan) is in place | | | |

IMS, incident management system; PHE, public health emergency; PHEOC, public health emergency operations centre.

**Table 3** PHEOC minimum requirements fulfilment by core component in Africa, 2021

| Core components of a PHEOC | Fully functional: PHEOCs that met ≥80% (≥15 of 19) # (%) | Functional: PHEOCs that met 60%–79% (11–14 of 19) # (%) | Partially functional: PHEOCs that met ≤50% (≤10 of 19) # (%) |
|---|---|---|---|
| PHEOC policies, plans and procedures | 10 (24) | 17 (34) | 14 (34) |
| Human resources, training and simulation exercises | 17 (41) | 7 (29) | 16 (39) |
| Information management and data standards | 24 (59) | 10 (24) | 7 (17) |
| Communication technology and physical infrastructure | 18 (44) | 11 (27) | 12 (29) |
| Total (n=41) | 12 (29) | 12 (29) | 17 (41) |

PHEOC, public health emergency operations centre.

Region MS in the WHO responded and 36 reported existence of a national PHEOC, 17 of these also had one at the subnational level.

Table 3 illustrates that 12 (29%) of the 41 national PHEOCs in the continent and 11 (31%) PHEOCs in the WHO African Region met 80% or more of the minimum requirements for the four core components of a PHEOC and classified as fully functional. In addition, 12 (29%) of the PHEOCs in the continent met 60%–79% and 17 (41%) met below 60% the minimum requirements and classified as functional and partially functional, respectively. The number of PHEOCs in the continent that meet the minimum requirements by PHEOC core components is displayed in table 3.

### Core component 1: PHEOC policies, plans and procedures

Twenty-five (61%) MS with PHEOCs have developed and enacted a legal framework to establish it, and 23 (56%) of the PHEOC legal frameworks have covered governance structure, core functions and the scope of authority (figure 1). In addition, 29 (71%) MS PHEOCs had a policy group to provide strategic guidance for the PHEOC operationalisation.

Twenty (49%) MS implemented a handbook for PHEOC operations and management to guide their day-to-day operations. Twenty-five (61%) have developed a national multihazard response plan that includes the concept of operations addressing priority risks and 19 (46%) of them were endorsed. In 24 (59%) MS with PHEOCs, a response plan covering various stakeholders' roles and responsibilities supporting response efforts was in place. Thirty-one (76%) of the PHEOCs had an IMS or comparable response structure and 25 (61%) could activate within 120 min as per JEE recommendation (table 4).

In the WHO African Region, 23 (56%) MS had a business continuity plan (BCP) and 18 (44%) had a handbook for PHEOC operations and management. In addition, 17 (41%) and 23 (56%) MS had an approved multihazard response plan and the capacity to activate their PHEOC within 120 min, respectively.

### Core component 2: human resources, training and simulation exercises

Thirty-eight (93%) of the PHEOCs indicated that at least one manager or focal point was present to oversee day-to-day operations and management of the PHEOCs and 26 (63%) had the minimum expected personnel (PHEOC manager, lead for operations, planning, logistics, finance and administration, communications officer, and ICT expert) to carry out routine preparedness activities. In 17 (41%) PHEOCs, staff were frequently oriented on PHEOC guides, and emergency management, and 14 (34%) PHEOCs regularly simulated exercises (SimEx) to test different capacities. When the IMS was activated, 32 (78%) PHEOCs could contact a roster of pretrained experts to assist with response coordination activities (table 5).

In the WHO African Region, 21 (51%) PHEOCs had the minimum expected routine staff, and 29 (71%) PHEOCs had staff trained on PHE preparedness and response.

### Core component 3: information management and data standards

There was a digital solution to process the data and information acquired from various sources in 33 (80%) of the PHEOCs. Data were routinely coming from relevant sectors (eg, surveillance and/or local health systems) in 32 (78%) PHEOCs. Thirty-five (85%) PHEOCs could collect and manage operational information (situational awareness) to inform action, whereas 26 (63%) PHEOCs were producing visual dashboards to convey a concise picture of the response and situation. In the WHO African Region, 31 (86%) PHEOCs could manage operational information to inform action and data were flowing systematically in 29 (81%) PHEOCs.

### Core component 4: communication technology and physical infrastructure

As indicated in figure 2, 27 (66%) of the PHEOCs had sufficient computer workstations, whereas 33 (80%) had workstations capable of serving the essential IMS

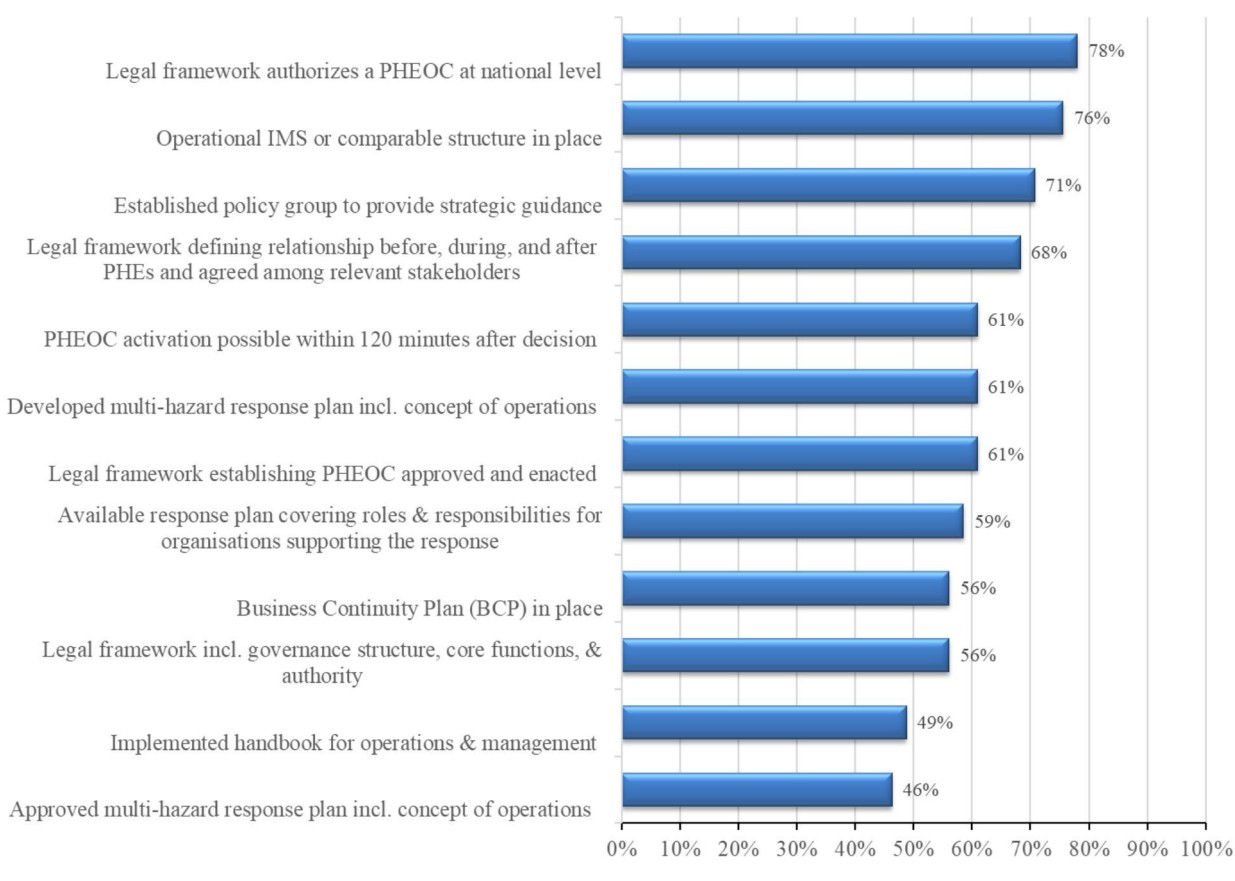

**Figure 1** PHEOCs with the capacity for core component 1 (PHEOC policies, plans and procedures) in Africa, 2021. IMS, incident management system; MS, Member States; PHEOCs, public health emergency operations centres.

personnel. Twenty-six (63%) had internet connectivity for the workstations and meeting rooms and 27 (66%) PHEOCs could hold a teleconference.

Twenty-eight (78%) and 21 (51%) PHEOCs with space for at least the IMS staff and adequate internet access, respectively, were from MS of the WHO African Region.

### DISCUSSION

The authors present findings on implementation status of PHEOCs at national levels in Africa from a survey conducted between May and November 2021 with the aim of assessing the progress made in PHEOC establishment and implementation status of the core components that make a PHEOC functional. In addition, it was also

**Table 4** Selected PHEOC capacities for core component 1 (PHEOC policies, plans and procedures) in Africa, 2021

| | PHEOCs | |
| --- | --- | --- |
| **PHEOC plans and procedures** | **No (#)** | **Proportion (%)** |
| Operational IMS or comparable structure in place | 31 | 76 |
| Developed multihazard response plan including the concept of operations | 25 | 61 |
| PHEOC activation possible within 120 min after decision | 25 | 61 |
| Available response plan covering roles and responsibilities of organisations supporting the response | 24 | 59 |
| Business continuity plan in place | 23 | 56 |
| Implemented handbook for PHEOCs operations and management | 20 | 49 |
| Approved multihazard response plan including concept of operations | 19 | 46 |

IMS, incident management system; PHEOC, public health emergency operations centre.

**Table 5** Selected PHEOC capacities for core component 2 (human resources and simulation exercises) in Africa, 2021

| Human resources, training and simulation exercises | PHEOCs | |
|---|---|---|
| | No (#) | Proportion (%) |
| Designated PHEOC manager | 38 | 93 |
| Staff trained on PHE preparedness and response | 33 | 80 |
| Contact a roster of trained personnel | 32 | 78 |
| Personnel to run response functions 24/7 | 26 | 63 |
| Minimum routine staff for IMS functions* | 26 | 63 |
| Dedicated training programme | 21 | 51 |
| Routinely trained staff on existing PHEOC guidance documents and exercises conducted to validate competencies | 17 | 41 |
| More than one exercise programme per year and documented after-action reviews | 14 | 34 |

*PHEOC manager and leads for operations, planning, logistics, finance and administration, communications officer, and ICT manager.
IMS, incident management system; PHE, public health emergency; PHEOC, public health emergency operations centre.

driven by the interest to have a resource document to facilitate evidence-based planning and support towards achieving functional PHEOCs in all MS in the African continent. The findings showed that the majority of the countries in the continent have designated a PHEOC either in a temporary or permanent facility. They further revealed that countries made efforts in implementing the four core components of a PHEOC including developing plans and procedures, train experts on PHEOC operations and IMS, strengthening information management and equipping a PHEOC with Information and Communication Technology (ICT) infrastructure. However, the implementation level of the core components varies from MS to MS.

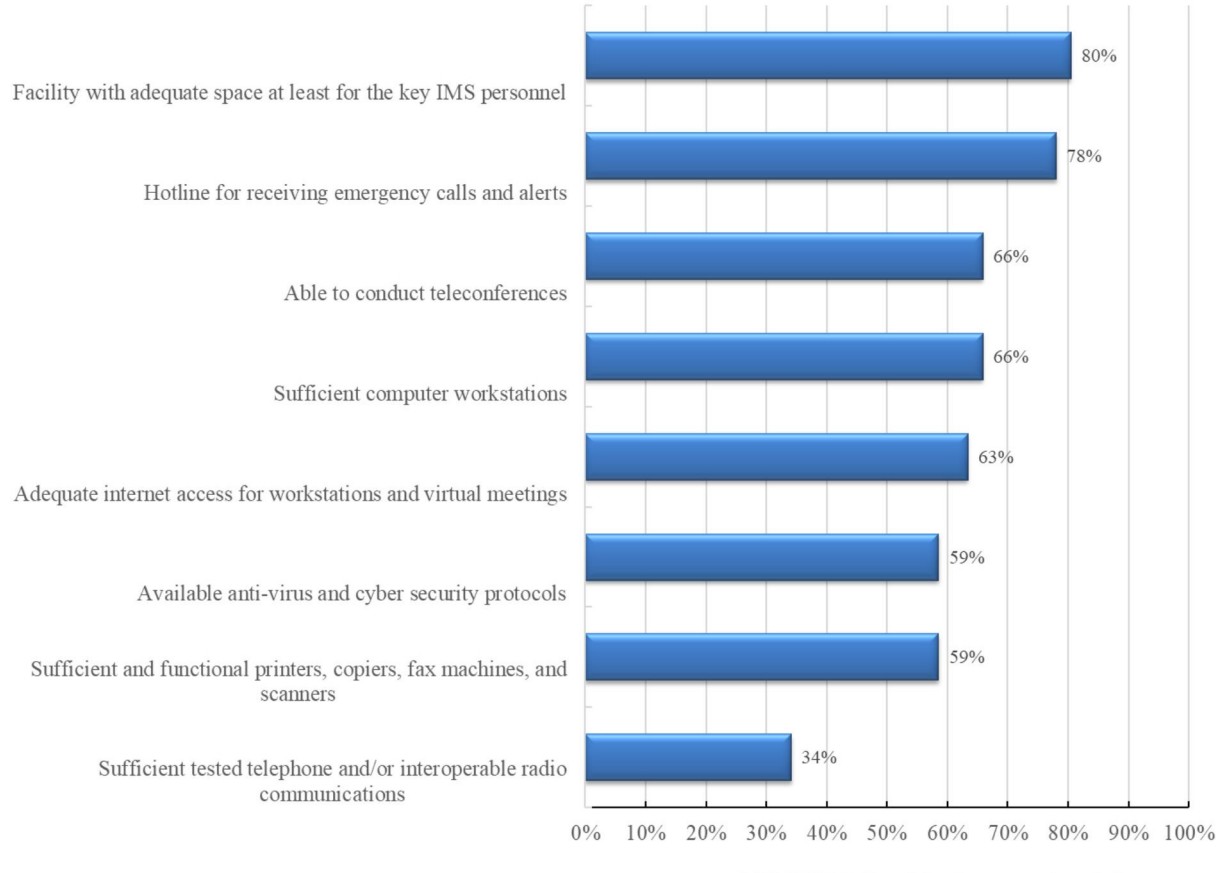

**Figure 2** PHEOCs with the capacity for core component 4 (communication technology and physical infrastructure) in Africa, 2021. IMS, incident management system; MS, Member States; PHEOCs, public health emergency operations centres.

According to the results, MS made significant progress as 41 (80%) of them established a PHEOC since the commencement of the regional initiative in 2015 that aimed to support MS with establishing PHEOCs to strengthen their emergency management capacity. This initiative was further intensified following the recommendations from the 2014 to 2016 EVD outbreak in West Africa and different stakeholders joining the effort to strengthen emergency preparedness and response capabilities of MS.[6] The results, however, showed that only 12 (29%) of the PHEOCs in the Africa met at least 80% of the minimum requirements and are fully functional. Furthermore, 12 (29%) and 17 (41%) of the PHEOCs in the continent met 60%–79% and below 60% the minimum requirements and classified as functional and partially functional, respectively. The capacity related to the basic indicators, however, differed. For example, in most of the MS, the core components of PHEOC policies, plans and procedures and the PHEOC workforce appear to have a critical gap, where only 10 (24%) and 17 (41%) met 80% and above of the minimum requirements. The main reason for this could be technical capacity, financial or other constraints. Most of the MS, on the other hand, had the required capability in terms of core component 3, information management and data standards.

Adverse health outcomes and economic disruption attributed to PHEs requires effective preparedness and response procedures.[2 10] A legal framework for approving the PHEOC's establishment and operation, a handbook for its operations and management, event or hazard-specific response and management plans, an incident action plan, and other pertinent plans and procedures must ideally be in place before any PHE occurs.[7] With respect to the core component of PHEOC policies, plans and procedures, only 20 PHEOCs had a handbook for PHEOC operations and management and a multihazard response plan including the concept of operations. In addition, a BCP was in place in only 23 PHEOCs. This might affect the effectiveness of the PHEOC to coordinate response during emergencies due to the absence of clear guidance and operational response plans.

Regarding the workforce capacity, despite most of the PHEOCs meeting the key requirements for some capabilities, only half of them had a dedicated training programme to train routine PHEOC and surge staff. Only 41% of MS PHEOCs had staff routinely trained on existing PHEOC guidance documents and conducted exercises to validate competencies. In addition, 34% of the PHEOCs had more than one exercise programme per year and implemented documented after-action reviews to address the gaps identified during exercises. Though the personnel needed for a PHEOC may vary due to many factors, including the emergency's size and complexity, a PHEOC usually has routine staff responsible for daily activities while surge personnel will be mobilised when the PHEOC is activated. The surge personnel, therefore, should be identified and trained in IMS and specific functions prior to PHE and an up-to-date roster should be maintained.[11] When the PHEOC is activated, it should be staffed with a team of subject matter experts drawn from the roster and regular training and exercises should be conducted to ensure the functioning, staffing and availability of a trained and skilled personnel.[11–15]

An EOC's lifeblood is information. Information management is a critical component of a functional PHEOC, it entails gathering, analysing, interpreting and distributing data promptly.[8–12 16] Event-specific data, event management information and context data are the three categories of data that must be consistently recorded, analysed, interpreted and displayed in a PHEOC.[7] During response coordination, a PHEOC requires specific types of data depending on the type of PHE. Data should be gathered according to local, that is, context-based and event-based conditions. During activation, EOCs use data technologies and informal networks of public health professionals to monitor epidemiological data and field reports from several sources.[13] In addition, the PHEOC should maintain clear and updated information about the incident or disaster. Effective communication is mandatory to retain the public's trust in messages and the function of the PHEOC.[17]

The majority of the PHEOCs had better performances for most parameters under the core component of communication technology and physical infrastructure. However, there were critical gaps in having interoperable radio communication and logistic facilities. The primary reason for this might be limitations in internet accessibility, funding and technical capabilities. This primarily affected the effectiveness of the PHEOC functionality in terms of communication ease, especially at the subnational level. Furthermore, the findings showed that most of the MS only have PHEOC capacity at the national level and there are important gaps at the subnational level, where most emergencies are managed. Addressing these gaps should urgently be considered in MS with lower emergency operations' capacity. The PHEOC should have the potential to acquire the technologies such as computers, phones, TV plasma screens, projectors and radio systems that support telecommunications, information management and visualisation of operational information resulting in more effective response coordination.[18]

## Limitation

The survey was self-administered, hence respondents did possibly not appraise performance status on the same scale. The largely binary nature of the questionnaire (yes/no options) may have limited the documentation of minor but important progress that was made. Last but not least, the survey does not deliver in-depth information for more detailed insight, as the main purpose was obtaining an overview of the PHEOC implementation status.

## CONCLUSION

WHO MS in Africa made significant progress in setting up PHEOCs to improve their emergency management

capabilities, the majority (80%) established a PHEOC since the regional PHEOC strengthening initiative started in 2015. Of these, one-fifth (12) MS have a national PHEOC fulfilling 80% or more of the minimum requirements to operate critical emergency functions and was classified as fully functional. The remaining have PHEOCs were classified as functional but needing improvement (29%) or partially functional (41%), respectively. PHEOCs in many MS have varying capabilities and need improvement to be fully functional and some MS (10) still have no PHEOCs to coordinate PHE response coordination.

The main bottlenecks for implementing functional PHEOCs meeting the requirements in all the four core components in Africa include the absence of a legal framework that clearly defines its mandate and functions, the lack of a standing policy group to provide operational support and strategic direction, unapproved plans and procedures, and limited availability of skilled human resources and funding for operations and sustainability.

This is the first in-depth analysis of PHEOC implementation status in Africa; the study was able to determine the implementation status of previously defined key capacities. The findings could be used for planning considerations in order to further improve PHE response capacities on the continent. It is crucial to enhance the implementation of functional PHEOCs according to the WHO framework and accompanying handbooks and guides. The implementation planning and execution processes require the involvement of all relevant stakeholders. In addition, regional and international partners need to support MS' efforts to address these gaps in developing functional PHEOC.

**Author affiliations**
[1]Emergency Preparedness and Response cluster, World Health Organization, Regional Office for Africa, Brazzaville, Congo
[2]Division of Emergency Preparedness and Response, African Centres for Disease Control and Prevention (Africa CDC), Addis Ababa, Ethiopia
[3]ECOWAS Regional Center for Surveillance and Disease Control, West African Health Organisation, Abuja, Nigeria
[4]WHO Health Emergencies Programme, World Health Organisation Regional Office for the Eastern Mediterranean, Cairo, Egypt
[5]WHO Health Emergencies Programme, World Health Organization, Geneva, Switzerland
[6]Centers for Disease Control and Prevention, Atlanta, Georgia, USA
[7]Bill & Melinda Gates Foundation, Seattle, Washington, USA
[8]Global Public Health Directorate, UK Health Security Agency, London, UK
[9]Department of infectious disease epidemiology, Robert Koch Institute, Berlin, Germany

**Acknowledgements** We thank all the MS in the continent for their significant contributions to the survey. We also want to thank everyone who helped review the survey tool and article.

**Contributors** STF, WM, WE, VL, YK, AA, LJ, MT, MCW, ER, VNB, EC, LM, AH, MA, INC, FB and ASG prepared the survey tool. STF and ALG conducted relevant article searches, developed a data analysis plan, analysed the data and wrote the manuscript. STF, ALG, WM, WE, VL, YK, AA, LJ, MT, MCW, ER, VNB, EC, LM, AH, MA, INC FB and ASG contributed to the writing and reviewed the final manuscript. STF is responsible for the overall content as the guarantor. All authors read and approved the final manuscript.

**Funding** The authors have not declared a specific grant for this research from any funding agency in the public, commercial or not-for-profit sectors.

**Disclaimer** The author is a staff member of the World Health Organization. The author alone is responsible for the views expressed in this publication and they do not necessarily represent the views, decisions or policies of the World Health Organization.

**Competing interests** None declared.

**Patient and public involvement** Patients and/or the public were not involved in the design, or conduct, or reporting, or dissemination plans of this research.

**Patient consent for publication** Not applicable.

**Provenance and peer review** Not commissioned; externally peer reviewed.

**Data availability statement** Data are available on reasonable request. Extra data can be accessed via the Dryad data repository at http://datadryad.org/ with the doi:10.5061/dryad.wm37pvmr9.

**ORCID iDs**
Senait Tekeste Fekadu http://orcid.org/0000-0002-8239-575X
Abrham Lilay Gebrewahid http://orcid.org/0000-0002-5570-5564
Ishata Nannie Conteh http://orcid.org/0000-0002-9200-189X

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
