## [Reviewer comments · BMJ Open]

ARTICLE DETAILS

TITLE (PROVISIONAL)	Public health emergency operations centres in Africa: a cross-sectional study assessing the implementation status of core components and areas for improvement, December 2021
AUTHORS	Fekadu, Senait; Gebrewahid, Abrham Lilay; Mankoula, Wessam; Eteng, Womi; Lokossou, Virgil; Kawe, Yan; Abdullah, Ali; Jian, L; Tut, Mathew; Wilton, Mencion Chuck; Rosenfeld, Emily; Bemo, Valerie; Collard, Emily; McGinley, Liz; Halm, Ariane; Aragaw, Merawi; Conteh, Ishata Nannie; Braka, Fiona; Gueye, Abdou Salam

VERSION 1 – REVIEW

REVIEWER	Huang, Yangmu Peking University Health Science Centre, School of Public Health
REVIEW RETURNED	19-Oct-2022

GENERAL COMMENTS	In general, this manuscript should be revised for English. There are errors and inconsistencies throughout the manuscript, and at times I struggled to understand the point being made without re-reading. 1. Introduction: the expression of text is not accurate enough, e.g., Monkeypox, and Yellow Fever should be in lowercase; no line should exist in "IHR-2005". The logic flow seems strange between Line 19 and 20. Were IHR requirements associated with the 2014-2016 ebola outbreak? In addition, some sentences are long to understand, e.g., the first and second sentences in paragraph 2.2. Methods: the self-assessment tool was not developed based on scientific basis. I can't see the clear relationship between the indicators and questions intended to answer. More details are needed to explain the expert consultation process, and how to keep your tool accuracy and standardized. Besides, I don't think Table 2 can be defined as indicators, as some of these are yes/no questions. What constitutes the calculation of requirements in percentage? Are there any differences among the weights of four core components and relevant "indicators"?3. Results: The overall results were clearly stated. But I haven't seen the clues to the results of each 19 "indicators". Legend and content are overlapped in Figure 2.4. Discussion: The first paragraph of discussion should mainly focus on concluding your results, instead of re-introducing the study objectives. The sentence in Line 53 and Line 56 are apparently repeated. For paragraph 3-6, it is very difficult to understand the point being made without re-reading.
---

REVIEWER	Bhuiyan, Mirza Mankweng Hospital, General surgery
-----------------	--

REVIEW RETURNED	04-Dec-2022
GENERAL COMMENTS	BMJ Open- Implementation status of public health emergency operations center in Africa, December 2021 Methodology Page 6, Line 21-22 What is the definition of minimum requirement of the core components for functional PHEOCs & what % will constitute minimum requirement? It is not clear (if >80% is optimum, then 51-79 and below 50 are what category! is it substandard). (Need to clear this issue) Result: Page 8; Line 3-8: in table 3 should be corrected according to methodology after defining the minimum requirement. Discussion: Page 9, Line 52-58 This sentence repeated twice in discussion and also in introduction. (A public health emergency operations center (PHEOC) is a hub designed to effectively coordinate information and resources during the management of PHEs) Discussion: Page 10, Line 10-11 PHEOC to strengthen their emergency management capability, from nearly zero in 2016 to 41 (80%) in 2021. Where is the reference for (from nearly zero in 2016) In conclusion (Page 11, line 31-32) it is also reflected (from nearly zero in 2016), so it need to be corrected before making conclusion of that statement. Abstract should be refreshed according to corrected main manuscript. Thanks Reviewer

VERSION 1 – AUTHOR RESPONSE

Reviewer: 1

Dr. Yangmu Huang, Peking University Health Science Centre

Comments to the Author:

In general, this manuscript should be revised for English. There are errors and inconsistencies throughout the manuscript, and at times I struggled to understand the point being made without re-reading.

1. Introduction:

the expression of text is not accurate enough, e.g., Monkeypox, and Yellow Fever should be in lowercase; no line should exist in "IHR-2005".

Addressed expression of texts and similar editing throughout the document.

The logic flow seems strange between Line 19 and 20. Were IHR requirements associated with the 2014-2016 ebola outbreak?

Sentences being made clear in the document – one of the recommendations that has been made after the 2014-2016 EVD outbreak was to improve capacities of MS to better prepare for and respond to future public health threats as required by IHR (2005).

Improving preparedness and response capacities of MS as required by IHR (2005) which requires State Parties to develop, strengthen and maintain their capacity to respond effectively to PHEs was

one of the recommendations that has been made following the EVD outbreak. In addition, some sentences are long to understand, e.g., the first and second sentences in paragraph 2.

Revised the sentences throughout the document and made an effort as much as possible to shorten the long sentences and make them clear to easily understand the key message.

2. Methods:

the self-assessment tool was not developed based on scientific basis. I can't see the clear relationship between the indicators and questions intended to answer.

The assessment tool is adapted from the WHO Framework for a public health emergency operations center. Annex 9 of the framework has a checklist to guide the implementation of a PHEOC or further improving an existing one. The assessment tool considered the capabilities across the four core components of a PHEOC describe in the framework and adapted to assess the implementation status of the PHEOCs in terms of the core components and identify the strengths (existing capacities in each component and capacities that are not still in place). As a result, the status of the PHEOCs was defined based on whether there have the capabilities or not under each core component.

More details are needed to explain the expert consultation process, and how to keep your tool accuracy and standardized.

As described the capabilities to assess the functionality of the PHEOCs were adapted from the PHEOC framework. However, the PHEOCs could not meet all the capabilities or requirements included in the tool. Experts represented from the key partners supporting PHEOC activities in the continent were consulted on what the minimum requirements/capabilities a PHEOC should meet. As a result, the experts considering their expertise and knowledge on the status of PHEOCs implementation in different countries in the continent held series of discussions and reached on consensus and selected the minimum requirements (19 as defined in Table 2) a PHEOC should meet to help accomplish critical emergency response functions.

Besides, I don't think Table 2 can be defined as indicators, as some of these are yes/no questions. It is true that the questions are yes/no and could be challenging to define the variables as indicators and agree to redefine them as capabilities.

What constitutes the calculation of requirements in percentage? Are there any differences among the weights of four core components and relevant "indicators"?

To calculate the percentage of countries with PHEOCs in the continent, it was calculated as: the number of countries that designated a PHEOC divided by the total number of countries in the African continent (n=54).

In addition, the percentages of PHEOCs that have the capability for each parameter/indicator were calculated as: the number of PHEOCs that has the capability/responded yes to the indicators under each core component divided by the total number of PHEOCs (both responded yes and no, n=41).

As most of the questions were yes/no, there was no difference in weights and gave same weight for all the core components and indicators.

3. Results:

The overall results were clearly stated. But I haven't seen the clues to the results of each 19 "indicators".

PHEOCs that have the capability for each parameter/indicator across the four core components of a PHEOC were described both in number and percentage (in description/narration and table). That's a description was provided to address how many of the PHEOCs had the capability (or responded, yes) for each parameter/indicator.

Legend and content are overlapped in Figure 2.

Yes – adjusted and new file uploaded.

4. Discussion:

The first paragraph of discussion should mainly focus on concluding your results, instead of re-introducing the study objectives.

Rewritten the first paragraph of the discussion and addressed as:

The findings showed that majority of the countries in the continent have designated a PHEOC either

in a temporary or permanent facility. The findings revealed that countries made efforts in implementing the four core components of a PHEOC including developing plans and procedures, train pool of experts on PHEOC operations and IMS, strengthening information management and designating and equipping a PHEOC facility with ICT infrastructure. However, the implementation level of the core components varies from MS to MS.

The sentence in Line 53 and Line 56 are apparently repeated.

Yes – the sentence in line 53 and line 56 are repeated and addressed as:

Both sentences are now deleted as they are not as such important and the first paragraph of the discussion need to be updated to address the comment.

For paragraph 3-6, it is very difficult to understand the point being made without re-reading.

Reviewed all paragraphs in the discussion and revised/edited as much as possible to shorten and make them clear to understand.

Reviewer: 2

Dr. Mirza Bhuiyan, Mankweng Hospital, University of Limpopo

Comments to the Author:

BMJ Open- Implementation status of public health emergency operations center in Africa, December 2021

Methodology

Page 6, Line 21-22

What is the definition of minimum requirement of the core components for functional PHEOCs & what % will constitute minimum requirement? It is not clear (if >80% is optimum, then 51-79 and below 50 are what category! is it substandard). (Need to clear this issue)

Revised the statement in the method section to clarify the classification of the PHEOCs based on the minimum capabilities required to run critical emergency response operations and help determine level of support for planning purpose. The statement was rephrased and addressed as:

The minimum capabilities required in a PHEOC to run the critical emergency response functions were selected from the parameters included in the survey. The selection of parameters was done based on the WHO framework for a PHEOC and expert consultation. As a result, 19 of the 31 (Table 2) parameters from the four core components of a PHEOC were prioritized and classified the PHEOCs into three categories for planning purpose. PHEOCs that meet 80% and above the minimum capabilities ($n \geq 15$ of the 19) labeled as fully functional, PHEOCs that meet 51%-79% ($n = 9$ to 14 of the 19) as functional but still need improvement, and PHEOCs that meet 50% and below ($n \leq 8$ of the 19) labeled as partially functional.

Result: Page 8; Line 3-8: in table 3 should be corrected according to methodology after defining the minimum requirement.

Corrected in the table as follows:

Fully functional: PHEOCs that met 80% and above the minimum capabilities ($n = 14$ of the 19),
Functional: PHEOCs that met 51%-79% the minimum capability parameters ($n = 9$ to 14 of the 19), and
Partially functional: PHEOCs that met 50% and below the minimum capability parameters ($n \leq 8$ of the 19)

Discussion: Page 9, Line 52-58

This sentence repeated twice in discussion and also in introduction. (A public health emergency operations center (PHEOC) is a hub designed to effectively coordinate information and resources during the management of PHEs)

Yes – the sentences are repeated. I kept the sentence in the introduction and deleted from the discussion as I believe it is not as such important to repeat it.

Discussion: Page 10, Line 10-11

PHEOC to strengthen their emergency management capability, from nearly zero in 2016 to 41 (80%) in 2021. Where is the reference for (from nearly zero in 2016)

The statement is now reworded to reflect that the establishment of PHEOC was initiated in 2015 after the inauguration of the regional initiative and was further intensified following the recommendations from the 2014-2016 EVD outbreak in West Africa. A reference also added and corrected as:

According to the findings, MS made significant progress and 41 (80%) of them established a PHEOC since the commencement of the regional initiative in 2015 and further intensified following the recommendations from the 2014-2016 EVD outbreak in West Africa.

In conclusion (Page 11, line 31-32) it is also reflected (from nearly zero in 2016), so it need to be corrected before making conclusion of that statement.

Addressed accordingly in the conclusion as follows:

Member States made significant progress in setting up a PHEOC to improve the emergency management capabilities 41 (80%) MS established a PHEOC since the commencement of the regional initiative in 2015.